# Validation of the NICHD Bronchopulmonary Dysplasia Outcome Estimator 2022 in a Quaternary Canadian NICU—A Single-Center Observational Study

**DOI:** 10.3390/jcm14030696

**Published:** 2025-01-22

**Authors:** Uthaya Kumaran Kanagaraj, Tapas Kulkarni, Eddie Kwan, Qian Zhang, Jeffery Bone, Sandesh Shivananda

**Affiliations:** 1Division of Neonatology, University of British Columbia and British Columbia Women’s Hospital, Vancouver, BC V6H 3N1, Canada; uthaya.kanagaraj@cw.bc.ca (U.K.K.); tapas.kulkarni@cw.bc.ca (T.K.); eddie.kwan@cw.bc.ca (E.K.); 2Department of Biostatistics, British Columbia Children’s Hospital Research Institute, Vancouver, BC V5Z 4H4, Canada; qian.zhang@bcchr.ca (Q.Z.); jbone@bcchr.ca (J.B.)

**Keywords:** bronchopulmonary dysplasia, preterm, neonates, validation, BPD estimator

## Abstract

**Background/Objectives**: The numerical risk of bronchopulmonary dysplasia (BPD) and/or death could be estimated using the National Institute of Child Health and Human Development (NICHD) BPD outcome estimator 2022 in extremely low gestational age (ELGA) infants during the first 4 weeks of life to facilitate prognostication, and center-specific targeted improvement interventions. However, the 2022 NICHD BPD outcome estimator’s performance in the Canadian setting has not been validated. Our objective is to validate the NICHD BPD outcome estimator 2022 in predicting death and or moderate to severe BPD at 36 weeks in infants less than 29 weeks admitted to NICU. **Methods**: A retrospective observational study (March 2022–August 2023) was conducted on both inborn and outborn preterm infants excluding neonates with major congenital anomalies. Infants were classified into six groups based on the predicted risk of death or Grade 2 or 3 BPD (<10%, 10–20%, 20–30%, 30–40%, 50–59%, ≥60%) followed by noting observed outcomes from the unit’s database. A receiver operating characteristics (ROC) curve was used to assess the accuracy of the NICHD BPD outcome estimator 2022, with an area under curve (AUC) > 0.7 defined a priori as an acceptable predictive accuracy for local use. **Results**: Among 99 infants included, 13 (13.1%) died, and 40 (40.4%) developed BPD. Median gestational age was 26 weeks, and median birth weight was 914 g. Twenty-three infants (23.2%) received postnatal steroids. The AUC values for death or moderate to severe BPD on days 1, 3, 7, 14, and 28 were 0.803, 0.806, 0.837, 0.832, and 0.843, respectively. The AUC values for moderate to severe BPD alone on those days were 0.766, 0.746, 0.785, 0.807, and 0.818 respectively. **Conclusions**: The 2022 BPD estimator adequately predicted the death and/or moderate to severe BPD on days 1, 3, 7, 14, and 28 of life. This tool could serve as a valid adjunct to facilitate discussion between clinicians and families on initiating time-sensitive targeted interventions to prevent or alter the course of BPD.

## 1. Introduction

Bronchopulmonary dysplasia (BPD) is a multifactorial disease affecting preterm infants, especially those who are born less than 29 weeks [1,2]. The incidence of BPD across Canadian centers varies between centers (20–60%) with a higher incidence in lower gestational age groups [2,3]. Infants with moderate to severe BPD are prone to developing chronic pulmonary hypertension (cPH), systemic hypertension, ventricular hypertrophy, and pulmonary vein stenosis (PVS) [4,5,6,7,8,9], and require prolonged ventilation, extended hospital stays, and home respiratory support [10]. After neonatal intensive care unit (NICU) discharge, these neonates face an increased risk of hospital readmissions within the first year, recurrent pulmonary infections, abnormal lung function [11,12,13], motor and cognitive impairments, behavioral and mental health challenges extending into childhood, and increased healthcare costs [14,15,16,17,18]. Clinicians use corticosteroids in preterm neonates to mitigate the inflammatory processes central to the pathogenesis of BPD [19]. However, the short-term benefits gained by steroids are often mitigated by long-term neurodevelopmental impairment (NDI) [20]. The literature suggests a risk-based approach to starting steroids, recommending treatment only for infants with a high risk of developing BPD [21,22,23,24,25]. This risk-based strategy aims to protect low-risk infants from potential harm because of steroid exposure, while ensuring that high-risk infants receive beneficial steroid therapy. However, implementing this approach in clinical practice is difficult because assessing the risk of BPD is highly subjective and varies significantly among clinicians. This is evident from the variations in steroid use for BPD observed both within and between Canadian neonatal networks (CNNs) centers [2,26].

The National Institute of Child Health and Human Development (NICHD) BPD outcome estimator 2022 is a web-based calculator that uses extensive population data from the Neonatal Research Network (NRN) to predict the likelihood of BPD or death at 36 weeks [27]. This tool aims to assist clinicians in making informed decisions early in the infant’s life. Although the tool is superior to the 2011 BPD estimator because of its sample size, contemporary cohort, and use of evidence-based definitions [27,28], Canadian NICUs have not validated it. Infants in Canadian NICUs were not part of the development cohort and may differ in disease severity, population diversity, and exposure to healthcare practices, and resource availability. The NICHD BPD outcome estimator 2022 uses the Jensen 2019 definition of BPD (based on a respiratory support at 36 weeks), while the CNN defines BPD based on both oxygen requirements and respiratory support at 36 weeks [2,29]. Locally, our quality improvement (QI) team sought to validate the generalizability of the NICHD BPD outcome estimator 2022 for local patient population before adopting it. We hypothesized that the NICHD BPD outcome estimator 2022 would accurately predict death and/or moderate to severe BPD, aiding clinicians in deciding early interventions and family counseling. This study aimed to validate the NICHD’s BPD outcome estimator 2022 in predicting death or Grade 2 or 3 BPD at 36 weeks with observed outcomes of death and/or moderate to severe BPD in infants born less than 29 weeks and admitted to British Columbia Women’s Hospital (BCWH).

## 2. Materials and Methods

This single-center retrospective observational study took place at BCWH, a quaternary-perinatal center in Canada. We included preterm inborn and outborn infants born at less than 29 weeks gestational age who were admitted to the NICU between March 2022 and August 2023. We excluded infants who had major congenital anomalies and those transferred to other units before reaching 36 weeks postmenstrual age (PMA). The NICU admission register was initially used to identify infants eligible for the study. Demographic and respiratory support data necessary for inputting into the NICHD BPD outcome estimator 2022 tool were collected from the hospital’s electronic health records (EHR). The CNN database, housed at BCWH, provided BPD, mortality, and resource utilization data. We estimated the risk of BPD (Grade 1, 2, and 3) and mortality using the NICHD BPD outcome estimator 2022 for the days 1, 3, 7, 14, and 28. We followed these selected time points in our study to ensure consistency with the validated NICHD 2022 online BPD outcome estimator tool. Whenever there was a doubt in ascertainment of BPD in the CNN database, e.g., nasal cannula flow rate, we reviewed the infant EHR to ensure accuracy in capturing observed outcomes. There were no major practice changes to alter BPD during the study period. We used oxygen saturation targets of 88 to 92%, minimally invasive surfactant treatment, and approved postnatal steroids such as systemic hydrocortisone to prevent BPD (STOP-BPD) [30], The Dexamethasone- A randomized trial (DART) [31], and inhaled budesonide regimens [32,33]. No postnatal steroid guidelines existed, and care teams decided by consensus.

We used the definitions of the NICHD BPD outcome estimator 2022 while interpreting the output, i.e., at 36 weeks PMA, Grade 1 BPD if requires 2 L/min nasal cannula or less, Grade 2 if requires more than 2 L/min nasal cannula or other forms of non-invasive ventilation support, and Grade 3 if requires invasive mechanical ventilation [27,29]. We used the observed severity of BPD using CNN database definition. At 36 weeks PMA, mild BPD if receives nasal cannula with 100% oxygen of <100 mL/min or 21–99% blended air/oxygen with flow rate of 1.5 L/min or less; moderate BPD if receives nasal cannula of 21–29% blended air/oxygen of 1.5 L/min or more, or 21–29% blended air/oxygen with continuous positive airway pressure (CPAP), synchronized positive airway pressure (SIPAP), non-invasive positive pressure ventilation (NIPPV) or non-invasive high frequency ventilation (NIHFV); severe BPD if 30% or more oxygen with nasal cannula of > 1.5 L/min or 30% of more of oxygen with CPAP, SIPAP, NIPPV, NIHFV or intubation and ventilation [34]. We validated the NICHD 2022 BPD outcome estimator using the CNN definition that’s already used in our unit. We believed that such alignment would facilitate local clinicians’ support for the tool’s adoption in clinical practice. To facilitate analysis of observed versus estimated outcomes, we classified infants into risk categories at time points day 1, 3, 7, 14, and 28, with an estimated risk of <10, 10–19, 20–29, 30–39, 40–49, 50–59%, and ≥60% from the NICHD BPD outcome estimator 2022. Next, we calculated the number of infants with the observed outcomes. We compared the estimator’s predicted probabilities with the observed outcomes. The unavailability of pre-study EHR data at our center hindered the collection of the NICHD’s BPD outcome estimator 2022 input variables, so we chose a convenience sample size. There was limited time available for validation prior to initiation of BPD prevention-related QI interventions locally. We used descriptive statistics to summarize baseline characteristics and NICU interventions. For infants who died before reaching 36 weeks PMA, and outborn infants with missing data for specific days before their admission to the NICU, we included them in the analysis on time points when their data were available. We generated receiver operating characteristic (ROC) curves to evaluate the predictive validity of the NICHD BPD outcome estimator 2022 at each time point. Using SPSS (Released 2023. IBM SPSS Statistics for Macintosh, Version 29.0.2.0 Armonk, NY, USA: IBM Corp.), we calculated the area under the curve (AUC) values. We selected an AUC greater than 0.7 for each ROC curve as the minimal acceptable predictive accuracy for local adoption of the BPD estimator tool [35]. This threshold denotes significant predictive accuracy. The hospital’s research ethics board approved the study (H24-00785).

## 3. Results

### 3.1. Baseline Characteristics

The study included 99 of the 118 infants who met the inclusion criteria, excluding 19 infants, as shown in Figure 1. On days 1, 3, 7, 14, and 28, there were 12, 12, 10, 14, and 17 infants’ data missing, but included in the analysis for those days when data were available. The median (interquartile range (IQR)) gestational age at birth was 26 (25–28) weeks, and the median (IQR) birth weight was 914 (755–1069) g. Table 1 presents other baseline characteristics. Appendix A show the interventions received by study infants during their NICU stay and the time these interventions lasted, respectively. Thirteen infants (13.1%) died, and by 36 weeks PMA, 40.4% developed moderate or severe BPD, with 32.3% and 7.1% having moderate and severe BPD, respectively. We show other outcomes at discharge in Table 2.

### 3.2. Respiratory Support

The median fractional inspired oxygen (FiO_2_) levels ranged from 21% to 26.5% at various time points over the study period. All infants received surfactant, of which 38% received it through minimally invasive surfactant technique (MIST). Of these, 16 (42.1%) were intubated and 6 (15.8%) received a second dose of surfactant via the endotracheal tube. We summarized other respiratory support details in Table 3. We present the observed and estimated predicted outcomes for moderate to severe BPD in Table 4 and for death or moderate to severe BPD in Table 5. The AUC values for moderate to severe BPD prediction on days 1, 3, 7, 14, and 28 were 0.766, 0.746, 0.785, 0.807, and 0.818, respectively (Table 4 and Figure 2). Similarly, the AUC values for death or moderate to severe BPD prediction on days 1, 3, 7, 14, and 28 were 0.803, 0.806, 0.837, 0.832, and 0.843, respectively (Table 5 and Figure 3). Comparison of ROC curves across days did not show a statistically significant difference (Appendix A).

## 4. Discussion

In this study, we found that the NICHD BPD outcome estimator 2022 adequately predicted the combined outcome of death and/or moderate to severe BPD on days 1, 3, 7, 14, and 28 of life. The accuracy improved with increasing postnatal age.

Of all the BPD prediction models developed in the past [36], two models—NICHD BPD outcome estimators 2011 and 2022 have been the most popular ones [27,28,36,37]. A major gap in the adoption of BPD prediction models has been the external validation in centers planning to adopt them. To address this gap, we evaluated the performance of the NICHD BPD outcome estimator 2022 in a Canadian setting.

External validation of the NICHD BPD outcome estimator 2011 accurately predicted death or severe BPD with an AUC of 0.81–0.84 [38], and 0.82 and 0.77 on Day 1 and 3 of life [39]. However, the model performed even better for predicting moderate to severe BPD alone with an AUC of 0.83 to 0.94 for Days 7 and 28 of life [39]. In the above studies, performance improved with increasing postnatal age. Two recent studies have externally evaluated the performance of the 2022 BPD estimator and its utility in predicting eventual steroid treatment [40,41]. Srivatsa et al.’s study found that the 2022 BPD estimator overestimated probabilities for babies who develop BPD, had low sensitivity to predict mortality, and overestimated postnatal steroid use [40]. A study by Kinkor et al. found that the NICHD BPD outcome estimator 2022 had poor to fair accuracy for predicting death or Grade 3 BPD on day 28 (AUC 0.77) but not at other time points in the first 28 days of life (AUC 0.58–0.67). However, the NICHD BPD outcome estimator 2022 showed good to excellent accuracy for identifying infants at high risk of steroid treatment (AUC 0.76–0.89) on Days 1, 3, 7, 14, and 28 [41]. The above two studies were retrospective, had small sample sizes, included infants admitted prior to 2021, had variable infant demographics, care practices (oxygen saturation targeting, postnatal steroid use), and BPD and mortality rates. Thus, validation in those studies is likely to have limited applicability in another center planning to adopt the estimator.

Based on our study findings, the NICHD BPD outcome estimator 2022 provides early and reliable prediction of infants at high risk of death and or moderate to severe BPD on days 1, 3, 7, 14, and 28. In our setting, we could use this estimator to consistently identify infants with a numerically high risk of death or moderate to severe BPD. We have reported increased variation in the timing of initiation of the first course of postnatal steroids (Median (IQR) of 28 (19–38) days) for prevention or treatment of BPD from our center [26]. Clinicians in our team believed the initiation of systemic steroids is not only dependent on the estimated risk of moderate or severe BPD or death but also on factors such as the infant’s age, comorbidities, and parental preferences. Consequently, we preemptively determined that any attempt at validating the 2022 BPD risk estimator against the initiation of systemic steroids in our unit would likely produce erroneous results. Therefore, we refrained from such analysis. Many studies in the past have suggested that earlier initiation of postnatal steroids ranging between 8 and 21 days of life was associated with decreased BPD compared with later initiation [42,43,44,45]. These studies along with this study’s results, suggest that using the NICHD BPD outcome estimator 2022 with its ability to provide a numerical risk of death or moderate to severe BPD might facilitate earlier conversations among clinicians and between clinicians and families on initiating targeted interventions to prevent or alter the course of BPD.

A lack of shared awareness regarding an individual infant’s predicted trajectory can hinder timely decision-making and prompt action, potentially delaying interventions that could reduce morbidity or the severity of conditions such as BPD [46]. Following the validation of the NICHD BPD outcome estimator 2022 tool, we aim to integrate it into our unit’s processes to enhance early recognition, foster situational awareness among the care team, and support shared decision-making. By doing so, we hope to implement more individualized, infant-centered care interventions that could reduce the incidence or severity of BPD. Specifically, we plan to screen all extremely preterm infants weekly to identify those at the highest risk for BPD. For these infants, we will initiate structured, facilitated interprofessional team huddles to establish and update a personalized weekly care plan, ensuring that care decisions are aligned with the evolving needs of each infant.

To our knowledge, this study is the first one to validate the NICHD BPD outcome estimator 2022 in a Canadian setting using a contemporary population. Using a pre-specified threshold value for acceptable performance of the model at our center is likely to make its use acceptable by clinicians and families. Additional strengths of this study include the use of a detailed dataset from a quaternary NICU and the application of the local CNN definition for BPD. Our study limitations include a single center with a small sample size, retrospective design, and exclusion of infants discharged prior to 36 weeks PMA. Although the above limitations may restrict the applicability of our findings, other NICUs could easily replicate our study methods to evaluate the 2022 NICHD BPD outcome estimator’s utility in their context.

## 5. Conclusions

We found that the NICHD BPD outcome estimator 2022 adequately predicted the death and/or moderate to severe BPD on days 1, 3, 7, 14, and 28 of life. These findings suggest clinicians could use it to identify infants at high risk of death or moderate to severe BPD during the first 28 days, facilitating discussions between clinicians and families about initiating targeted interventions to prevent or alter the course of BPD and engaging families in treatment-related risk versus benefit conversations.

## Figures and Tables

**Figure 1 jcm-14-00696-f001:**
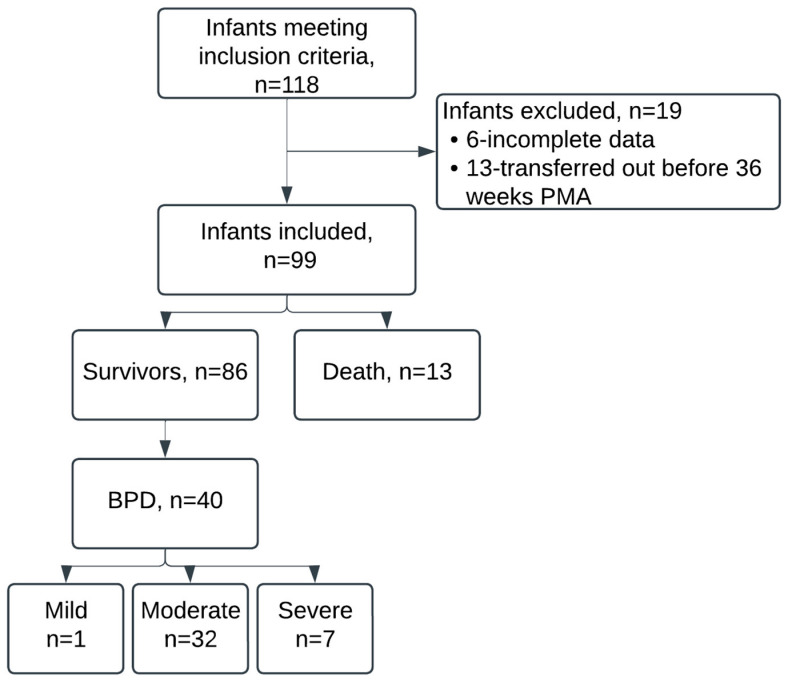
Study flow diagram.

**Figure 2 jcm-14-00696-f002:**
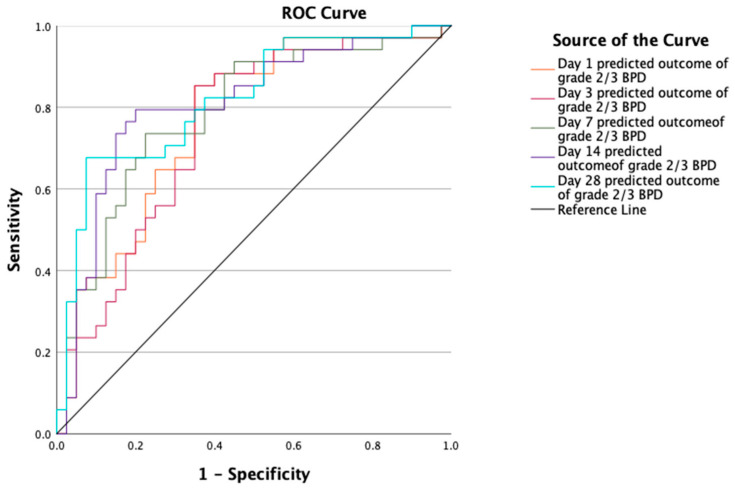
Receiver operating characteristic (ROC) curve for estimated predictive accuracy of moderate to severe BPD.

**Figure 3 jcm-14-00696-f003:**
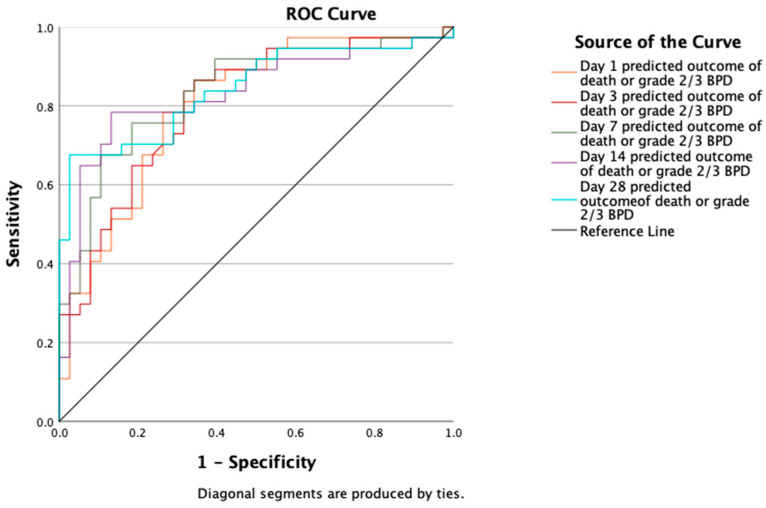
Receiver operating characteristic (ROC) curve for estimated predictive accuracy of death or moderate to severe BPD.

**Table 1 jcm-14-00696-t001:** Demographic profile of infants at admission.

Variable	Total Infants (99)
Gestational age at birth, weeks, median (IQR)	26 (25–28)
22–24 weeks, median (IQR)	18 (18.2)
25–26 weeks, median (IQR)	38 (38.4)
27–28 weeks, median (IQR)	43 (43.4)
Birth weight, grams, median (IQR)	914 (755–1069)
Male	57 (57.6)
Outborn	22 (22.2)
Antenatal betamethasone (partial or complete)	86 (86.9)
Suspected chorioamnionitis	24 (24.2)
C-Section delivery	68 (68.7)
APGAR 1 min, median (IQR)	5 (2–6)
APGAR 5 min, median (IQR)	6 (6–8)
Intubation and ventilation during resuscitation	62 (62.6)
SNAPPE II ^1^	27 (18–42)
SNAPPE II ^1^ > 20	60 (60.6)

^1^ SNAPPE II—Score for Neonatal Acute Physiology Perinatal Extension II. Value in cell n (%) unless specified, IQR—Inter quartile range

**Table 2 jcm-14-00696-t002:** Outcome of infants at discharge from NICU.

Variable	n (%)
Mortality	13 (13.1)
BPD	40 (40.4)
Moderate BPD	32 (32.3)
Severe BPD	7 (7.1)
ROP Stage ≥ 3 right or left	33 (33.3)
ROP treated, right or left	8 (8.8)
PDA treated (medical or surgical)	42 (42.4)
Pneumothorax	5 (5.1)
Surgical NEC	14 (14.1)
IVH Grade ≥ 3	29 (29.3)
PVL Grade > 2	9 (9.1)
Spontaneous intestinal perforation	3 (3.0)
Discharge—survival without major morbidity ^1^	25 (25.3)
Discharge oxygen	2 (2.3)
Discharge monitor	46 (53.5)
Discharge ostomy ^2^	0 (0)
Discharge gavage	37 (43.0)
Discharge tracheostomy	1 (1.2)
Discharge gastrostomy	11 (12.8)
Discharge non-invasive ventilation	3 (3.5)
Discharge continuous positive airway pressure	3 (3.5)
Discharge—technology dependency ^3^	55 (64.0)

NICU—neonatal intensive care unit, BPD—bronchopulmonary dysplasia, ROP—retinopathy of prematurity, PDA—patent ductus arteriosus, NEC—necrotizing enterocolitis, IVH—interventricular hemorrhage, PVL—periventricular leukomalacia. ^1^ Moderate or severe BPD, IVH ≥ Gr 3/PVL, NEC ≥ 2, ROP Stage ≥ 3/treatment, nosocomial infection as per CNN definition [2]; ^2^ ileostomy or colostomy (not including tracheostomy or gastrostomy) at the time of discharge or transfer; ^3^ any of the following oxygen, monitor, gavage, tracheostomy, gastrostomy, ventilation, or continuous positive airway pressure. The denominator is 86 infant survivors for discharge-related variables.

**Table 3 jcm-14-00696-t003:** Respiratory support received by infants at various time points during NICU stay.

Variable		Day 1	Day 3	Day 7	Day 14	Day 28	At 36 Weeks PMA
Fio_2_ ≥ 22%		37/87 (42.5)	49/87 (56.3)	54/89 (60.7)	57/85 (67.1)	48/82 (58.5)	15/86 (17.4)
Fio_2_, median (IQR)		21 (21, 25.75)	23 (21, 26)	23 (21, 30)	26.5 (21, 34)	24 (21, 30)	21 (21, 25.5)
Respiratory support	HFV	12/87 (13.8)	17/87 (19.5)	24/89 (27)	20/85 (23.5)	15/82 (18.3)	5/86 (5.8)
	CMV	43/87 (49.4)	29/87 (33.3)	17/89 (19.1)	13/85 (15.2)	6/82 (7.3)	2/86 (2.3)
	NIPPV	7/87 (8)	22/87 (25.3)	25/89 (28.1)	22/85 (25.9)	30/82 (36.6)	4/86 (4.6)
	CPAP	24/87 (27.6)	18/87 (20.7)	22/89 (24.7)	28/85 (33)	16/82 (19.5)	13/86 (15.1)
	HFNC	0	0	0	1/85 (1.2)	11/82 (13.4)	15/86 (17.4)
	LFNC	0	0	0	0	0	1/86 (1.2)
No respiratory support		1/87 (1.2)	1/87 (1.2)	1/89 (1.1)	1/85 (1.2)	4/82 (4.9)	46/86 (53.5)

Each cell has numerator as number of infants receiving intervention on those days and denominator as number of infants; (%) unless otherwise specified; Fio_2_—fractional inspired oxygen, IQR -inter quartile range, HFV—high frequency ventilation, CMV—conventional mechanical ventilation—as volume targeted ventilation, NIPPV—non-invasive positive pressure ventilation, CPAP—continuous positive airway pressure, HFNC—high-flow nasal cannula (>1.5 L/min), LFNC—low-flow nasal cannula (<1.5 L/min).

**Table 4 jcm-14-00696-t004:** Observed incidence of moderate to severe BPD stratified by pre-specified time points.

	Estimated Risk of Grade 2/3 BPD Using the Calculator							AUC	95% CI for AUC
	<10%	10–19%	20–29%	30–39%	40–49%	50–60%	≥60%		
Day 1 (n = 87)	4/25 (16%)	7/20 (35%)	8/17 (47.1%)	11/17 (64.7%)	4/7 (57.1%)	0/1	0/0	0.766	0.657–0.875
Day 3 (n = 87)	2/24 (8.3%)	8/17 (47.1%)	10/20 (50%)	5/11 (45.5%)	2/6 (33.3%)	7/9 (77.8%)	0/0	0.746	0.633–0.860
Day 7 (n = 89)	3/25 (12%)	6/20 (30%)	6/10 (60%)	7/11 (63.6%)	10/18 (55.6%)	3/5 (60%)	0/0	0.785	0.678–0.891
Day 14 (n = 85)	3/23 (13%)	9/26 (34.6%)	2/3 (66.7%)	7/10 (70%)	4/10 (40%)	8/9 (88.9%)	¾ (75%)	0.807	0.703–0.911
Day 28 (n = 82)	2/20 (10%)	9/29 (31%)	9/12 (75%)	2/2 (100%)	½ (50%)	6/7 (85.7%)	9/10 (90%)	0.818	0.720–0.916

Each cell shows the observed outcome (moderate to severe BPD) as the numerator and the predicted outcome from the BPD estimator at a pre-specified time point as the denominator. BPD—bronchopulmonary dysplasia, AUC—area under the curve, CI—confidence interval.

**Table 5 jcm-14-00696-t005:** Observed incidence of death or moderate to severe BPD stratified by pre-specified time points.

	Estimated Risk of Death or Grade 2/3 BPD Using the Calculator							AUC	95% CI for AUC
	<10%	10–19%	20–29%	30–39%	40–49%	50–60%	≥60%		
Day 1 (n = 87)	2/21 (9.5%)	5/15 (33.3%)	9/14 (64.3%)	9/12 (75%)	7/11 (63.6%)	5/5 (100%)	8/9 (89%)	0.803	0.703–0.903
Day 3 (n = 87)	2/23 (8.7%)	6/16 (37.5%)	9/14 (64.3%)	7/10 (70%)	6/8 (75%)	4/6 (66.7%)	10/10 (100%)	0.806	0.707–0.905
Day 7 (n = 89)	2/19 (10.5%)	6/19 (31.6%)	6/13 (46.2%)	8/10 (80%)	6/7 (85.7%)	6/8 (75%)	13/13 (100%)	0.837	0.745–0.930
Day 14 (n = 85)	2/20 (10%)	5/23 (21.7%)	5/7 (71.4%)	4/6 (66.7%)	9/10 (90%)	7/9 (77.8%)	9/10 (90%)	0.832	0.734–0.930
Day 28 (n = 82)	2/20 (10%)	9/27 (33.3%)	8/12 (66.7%)	2/3 (66.7%)	2/2 (100%)	2/2 (100%)	15/16 (93.8%)	0.843	0.751–0.935

Each cell shows the observed outcome (death or moderate to severe BPD) as the numerator and the predicted outcome from the BPD estimator at a pre-specified time point as the denominator. BPD—bronchopulmonary dysplasia, AUC—area under the curve, CI—confidence interval.

## Data Availability

The data presented in this study are available on request from the corresponding author because of restrictions from the hospital on sharing individual patient data due to privacy concerns, and as per the original approval of secondary use of data housed in the Canadian Neonatal Network database.

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
