# Peer review of "Validation of the NICHD Bronchopulmonary Dysplasia Outcome Estimator 2022 in a Quaternary Canadian NICU—A Single-Center Observational Study"

_jcm, 2025, doi:10.3390/jcm14030696_

Round 1
Reviewer 1 Report
Comments and Suggestions for Authors
In the paper Validation of the NICHD Bronchopulmonary Dysplasia Outcome Estimator 2022 in a Quaternary Canadian NICU – a single centre observational study, the authors aimed to validate the NICHD BPD outcome estimator 2022 in predicting death and or moderate to severe BPD at 36 weeks through a retrospective observational study on a cohort of preterm infants less than 29 weeks admitted to NICU.
Overall, the article is well written and could be useful for clinicians involved in the care of extremely preterm infants, provided future studies further validate the role of this tool.
The authors reported a diverse palette of strategies for initiation and regimens of postnatal steroids utilized in their centre. However, after validating the NICHD BPD outcome estimator 2022 in consistently identifying infants with a numerically high risk of poor outcomes, they stated that they “…could not validate the utility of NICHD BPD outcome estimator 2022 in identifying infants at high risk of steroid treatment.”
My suggestion for the authors is to rephrase the paragraph between the lines 229 and 236 (which is too vague) and to try to clarify how they envision the utility of this tool in the clinician’s decision to choose a strategy to prevent or alter the course of BPD in high-risk infants, as this would prove its usefulness in clinical setting.

Reviewer 2 Report
Comments and Suggestions for Authors
Please find attached
